# Good Meta-tasks Make A Better Cross-lingual Meta-transfer Learning for Low-resource Languages

**Linjuan Wu[1], Zongyi Guo[2,3], Baoliang Cui[2,3], Haihong Tang[2,3], Weiming Lu[1,†]**

[1]College of Computer Science and Technology, Zhejiang University
[2]Alibaba International Digital Commerce Group, China
[3]Alibaba Group, China
[1]{wulinjuan525,luwm}@zju.edu.cn
[2,3]{zongyi.gzy, moqing.cbl, piaoxue}@alibaba-inc.com

## Abstract

Model-agnostic meta-learning has garnered attention as a promising technique for enhancing few-shot cross-lingual transfer learning in low-resource scenarios. However, little attention was paid to the impact of data selection strategies on this cross-lingual meta-transfer method, particularly the sampling of cross-lingual meta-training data (i.e. meta-tasks) at the syntactic level to reduce language gaps. In this paper, we propose a **Me**ta-**Ta**sk **Co**llector-based **Cross**-lingual **M**eta-**T**ransfer framework (MeTaCo-XMT) to adapt different data selection strategies to construct meta-tasks for meta-transfer learning. Syntactic differences have an effect on transfer performance, so we consider a syntactic similarity sampling strategy and propose a syntactic distance metric model consisting of a syntactic encoder block based on the pre-trained model and a distance metric block using Word Move's Distance (WMD). Additionally, we conduct experiments with three different data selection strategies to instantiate our framework and analyze their performance impact. Experimental results on two multilingual NLP datasets, WikiAnn and TydiQA, demonstrate the significant superiority of our approach compared to existing strong baselines[1].

## 1 Introduction

Few-shot cross-lingual transfer surpasses zero-shot transfer (Lauscher et al., 2020; Hu et al., 2020; Zhao et al., 2021) using multilingual pre-trained language models (PLMs) (Devlin et al., 2019; Pires et al., 2019; Conneau and Lample, 2019; Conneau et al., 2020; Chi et al., 2021). It significantly improves model performance in the target language with minimal annotation costs. Recent studies have highlighted the benefits of Model-Agnostic Meta-Learning (MAML) (Finn et al., 2017) for few-shot cross-lingual transfer learning in NLP

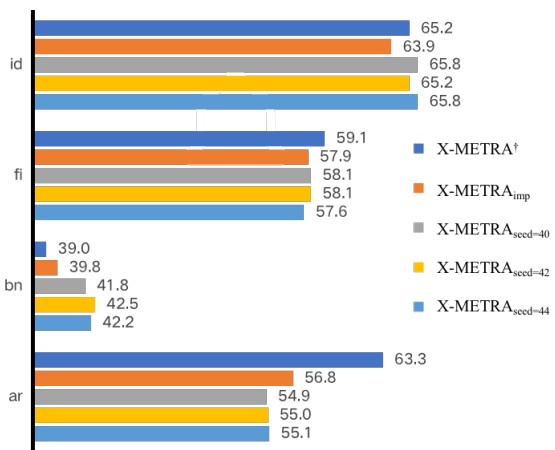

Figure 1: EM score of 4 languages on the TydiQA dataset based on the X-METRA model. † refers to results from (M'hamdi et al., 2021). The subscript *imp* means the result from our implementation. The *seed* indicates the random seed number for random sampling.

tasks (Nooralahzadeh et al., 2020; Ponti et al., 2021; Liu et al., 2021; M'hamdi et al., 2021). Our focus in this paper is primarily on cross-lingual meta-transfer methods for low-resource languages (Liu et al., 2021; M'hamdi et al., 2021), utilizing support sets from high-resource languages to establish an effective initialization for training in the low-resource target language.

The selection strategies for meta-tasks in cross-lingual meta-transfer learning typically involve random sampling (Nooralahzadeh et al., 2020; Ponti et al., 2021; Liu et al., 2021) or semantic similarity (Wu et al., 2020a; M'hamdi et al., 2021). We compare these two strategies using the X-METRA model (M'hamdi et al., 2021) on four languages from the TydiQA dataset (Clark et al., 2020). Figure 1 illustrates that models trained with randomly sampled meta-tasks (X-METRA_{seed}) in different seed setting generally perform a big variance of results compared with X-METRA_{imp}. Different random seed means the different selection of in-

---

†Corresponding authors.

[1]The code is available at https://github.com/wulinjuan/MeTaCo-XMT

stances to construct meta-task, and better selection can generate better results. It emphasize the pivotal role of meta-task construction in enhancing cross-lingual meta-transfer learning.

Motivated by the above finding, we propose a **Meta-Ta**sk **Co**llector-based **Cross**-lingual **M**eta-**T**ransfer framework (MeTaCo-XMT). As shown in Figure 2(a), the meta-task collector includes a data encoder and data selector with a distance metric block, which can be designed according to different data selection strategies. For instance, multilingual semantic representation and cosine similarity metrics can realize semantic similarity sampling strategy. Specifically, the data encoder encodes all data to semantic space, and the data selector selects top-v cosine-similar support instances for each query data as candidates. Finally, the meta-task can be constructed according to the setting.

However, the semantic similarity sampling is limited for different tasks. For example, in the machine reading comprehension (MRC) task, most semantically similar samples are subject-related, ignoring the relationship between question and paragraph. In practice, different questions with a common paragraph may choose the same support instance. For structural tasks like named entity recognition (NER), syntactic similarity sampling is more effective due to language-specific syntactic differences. Therefore, we propose a Syntactic Distance Metric Model (SDMM) based on multilingual PLM and the Word Mover's Distance (WMD) (Kusner et al., 2015). As shown in Figure 2(b), the SDMM incorporates a syntactic linear layer for syntactic tree learning and employs a triplet loss to distinguish WMD between close and distant languages from pivot languages (e.g., English).

In order to further explore the impact of data selection strategy on cross-lingual meta-transfer performance, we compare three sampling strategies to build meta-tasks including semantic similarity sampling, task-similarity sampling, and syntactic similarity sampling. We conduct experiments on 13 typologically diverse target languages of two cross-lingual tasks: NRE and MRC. Our main contributions are listed below:

- We propose a meta-task collector-based cross-lingual meta-transfer framework (MeTaCo-XMT) to accommodate different data select strategies to reducing the gap of languages.

- We propose a syntactic distance metric model to calculate the distance of text pairs at the syntactic level for syntactic similarity sampling.

- We investigate three different data selection strategies and experiment on two cross-lingual datasets (WikiAnn and TydiQA) to demonstrate that our framework equipped with syntactic similarity sampling strategy significantly outperforms existing strong baselines.

## 2  Related Work

We focus on two threads of related work: (1) meta-learning for cross-lingual transfer and (2) training data selection. Sherborne and Lapata (2023) and Wu et al. (2020b) use meta-learning for cross-lingual NER and Semantic Parsing with a slight enhancement in minimal resources. X-MAML (Nooralahzadeh et al., 2020) combines the MAML and cross-lingual transfer method based on PLM and demonstrates improvement in zero-shot and few-shot settings. X-MAML samples the support and query data from the same language, which limits the ability of the model for cross-lingual transfer. XLA-MAML (Liu et al., 2021) performs direct cross-lingual adaptation in the meta-learning stage by sampling the meta-tasks from two or more languages. X-METRA-ADA (M'hamdi et al., 2021) follows the setting of cross-lingual meta-transfer training and adds a meta-adaptation stage for further improvement. While Wu et al. (2020b) and M'hamdi et al. (2021) use semantic similarity to select the meta-tasks, we explore more data selection strategies to get better cross-lingual meta-transfer performance.

Training data selection has been extensively studied for several tasks, such as domain adaptation (Liu et al., 2019; Ivison et al., 2022) and cross-lingual transfer (Maurya and Desarkar, 2022; Kumar et al., 2022). In the cross-lingual transfer setting, the selection of training data can reduce the performance gaps across languages. Kumar et al. (2022) proposed approaches of data selection rely on multiple measures such as data entropy using an n-gram language model, predictive entropy, and gradient embedding. Maurya and Desarkar (2022) use meta-learning techniques for cross-lingual generation and choose centroid languages to meta-training the model and improve other languages. In this paper, we propose a cross-lingual meta-transfer framework based on a meta-task collector and explore the performance of different data sampling strategies.

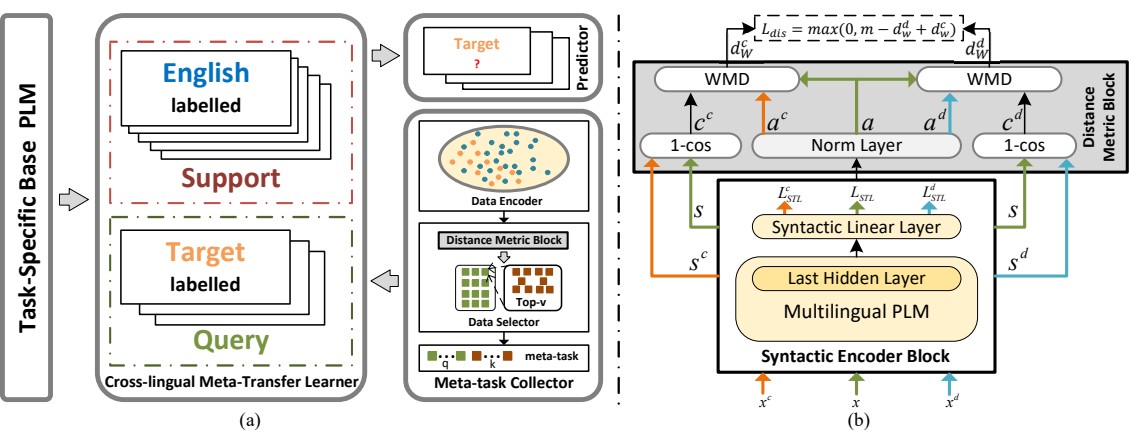

Figure 2: Diagram of (a) the proposed meta-task collector-based cross-lingual meta-transfer framework and (b) the syntactic distance metric model trained with triplet loss $L_{dis}$ and syntax tree loss $L_{STL}$. The superscript $d$ and $c$ in (b) represent the distant and close language from pivot language (without superscript) respectively.

## 3 Preliminaries

**Model Agnostic Meta-Learning** (MAML) (Finn et al., 2017) as an optimization-based meta-learning method is compatible with any model $f_\theta$ that updates parameters $\theta$ through gradient descent. Formally, given a task $\tau_i$ from meta-tasks set $\tau$ with loss function $\mathcal{L}_{\tau_i}$, the optimal target of MAML is:

$$\theta = \arg\min_\theta \sum_{\tau_i \in \tau} \mathcal{L}_{\tau_i}(f_{\theta_i'}), \qquad (1)$$

$$\theta_i' = \theta - \alpha\nabla_\theta\mathcal{L}_{\tau_i}(f_\theta), \qquad (2)$$

where the $\tau_i = \{\mathcal{S}_i, \mathcal{Q}_i\}$, consisting of a support set $\mathcal{S}_i$ and a query set $\mathcal{Q}_i$. MAML optimizes the parameters toward the optimal target in Equation 1 via an inner step and a meta step. The support set $\mathcal{S}_i$ is used for the inner step (as Equation 2) to update the $\theta$ to $\theta'$ with learning rate $\alpha$. The meta step uses the updated model $f_{\theta_i'}$ and optimizes $\theta$ with a learning rate $\beta$ across query set $\mathcal{Q}_i$, as follow:

$$\theta \leftarrow \theta - \beta\nabla_\theta\mathcal{L}_{\tau_i}(f_{\theta_i'}) \qquad (3)$$

The optimized-based meta-learning algorithm can perform the model fast adaptation to a new task during the adaptation training phase.

**Cross-lingual Meta-transfer Learning** mainly focuses on the meta-training phase of MAML. In the inner step, the model $f_\theta$ learns a good initialization of parameter $\theta'$ by repeatedly simulating the learning process on source language support set $\mathcal{S} \in \mathbb{D}_{src}$. The meta step is fine-tuning the initialized model in a target language query set $\mathcal{Q} \in \mathbb{D}_{tgt}$. These two steps are iterated repeatedly to optimize the parameters $\theta$. We skip the adaptation phase and directly evaluate the target language test dataset.

## 4 Methodology

Figure 2 shows the architecture of our meta-task collector-based cross-lingual meta-transfer framework and the proposed syntactic distance metric model. Our framework consists of three essential components: task-specific base PLM, cross-lingual meta-transfer learner, and meta-task collector. In this section, we first introduce the training procedure of MeTaCo-XMT (§ 4.1) and a detailed description of the meta-task collector (§ 4.2). Then syntactic distance metric model (§ 4.3) is proposed to instantiate the meta-task collector and two other data selection strategies are described in § 4.4.

### 4.1 Training Procedure of MeTaCo-XMT

As shown in Figure 2(a), for the task-specific base PLM, we first initialize our model $f_\theta$ with multilingual PLM such as mBERT (Pires et al., 2019) or XLM-R (Conneau et al., 2020) and fine-tune it on English monolingual labeled data. This step allows the PLM to take benefit of the high resource data and to serve as a baseline model.

Before the cross-lingual meta-transfer stage, we sample a batch of meta tasks $\mathcal{T} = \{\mathcal{S}, \mathcal{Q}\}$ from dataset $\mathbb{D} = \{\mathbb{D}_{src}, \mathbb{D}_{tgt}\}$ by meta-task collector, which uses high-resource source language (typically English) data in support sets and low-resource target language data in the query sets. For every task $\tau_i = \{\mathcal{S}_i, \mathcal{Q}_i\}$, we update $\theta_i'$ over $r$ steps using support instances in $\mathcal{S}_i$, as Equation 2. At the end of inner loop, we compute the gradients with respect to the loss of $\theta_i'$ on $\mathcal{Q}_i$. After each batch training, we sum over all pre-computed gradients and update $\theta$, thus completing one outer loop.

At test stage, we directly evaluate the optimized

model $f_\theta$ on the target language test dataset.

## 4.2 Meta-Task Collector

The large variations illustrated in Figure 1 demonstrate that meta-task selection has a profound impact on the performance of cross-lingual meta-transfer learning. So we equip the cross-lingual meta-transfer framework with a meta-task collector to select the training data strategically. As shown in Figure 2(a), our meta-task collector is to construct a meta-task with $q$ query instances and $k$ support instances, described in the following.

In this few-shot cross-lingual transfer setting, we have a rich-resource source language (i.e. English) dataset $\mathbb{D}_{src} = \{(x_{src}, y_{src})\}$ and a low-resource target language dataset $\mathbb{D}_{tgt} = \{(x_{tgt}, y_{tgt})\}$, where $(x, y)$ is a pair of text $x$ and ground truth labels $y$. We form the meta-tasks $\mathcal{T} = \{\mathcal{S}, \mathcal{Q}\}$ with the following process:

1. Sample $num$ target data as query instances set $\mathbb{Q} = \{(x_{tgt}^{(i)}, y_{tgt}^{(i)})\}_{i=1}^{num} \in \mathbb{D}_{tgt}$. The entire source dataset is used as candidate support instances $\mathbb{S} = \{(x_{srs}^{(j)}, y_{src}^{(j)})\}_{j=1}^{M} \in \mathbb{D}_{src}$.

2. Encode text $x$ in query and support instances set by encoder block to obtain vector $\boldsymbol{s}$.

3. Calculate the distance $d_{ij}$ between $\boldsymbol{s}_{tgt}^{(i)}$ and $\boldsymbol{s}_{src}^{(j)}$ by distance metric block (DMB):

$$d_{ij} = \mathrm{DMB}(\boldsymbol{s}_{tgt}^{(i)}, \boldsymbol{s}_{src}^{(j)}), \quad (4)$$

For each query instance $q_i = (x_{tgt}^{(i)}, y_{tgt}^{(i)})$, we choose top-v closest instances as its candidate support subset $\mathbb{S}_{q_i} = \{(x_{srs}^{(ij)}, y_{src}^{(ij)})\}_{j=1}^{v} \in \mathbb{S}$.

4. Finally, we draw a task $\tau_i = \{\mathcal{S}_i, \mathcal{Q}_i\} \in \mathcal{T}$ by first randomly choosing $q$ query instances, forming $\mathcal{Q}_i$. For each query instance $q_j$ in $\mathcal{Q}_i$, we draw the $k/q$ most closest candidate support instance from $\mathbb{S}_{q_j}$ thus forming $\mathcal{S}_i$. The number of meta-tasks is also a hyperparameter.

How to adapt the meta-task collector to syntactic similarity sampling is described in detail below.

## 4.3 Syntactic Distance Metric Model

Linguistic disparities affect the performance of cross-lingual transfer (Pires et al., 2019; K et al., 2020; Wu et al., 2022), such as the difference in word order or other syntactic differences. So

we propose a Syntactic Distance Metric Model (SDMM) to select syntactic-similar source instances for meta-learning, which consists of a syntactic encoder block and a distance metric block.

As shown in Figure 2(b), We first encode the multilingual text into a universal syntactic space and use Word Mover's Distance (WMD) (Kusner et al., 2015) to measure the syntactic distance of text pairs. To train the model, we optimize a triplet loss by using three-way parallel texts $(x, x^d, x^c)$. The three languages include the pivot language (English), a distant language $l^d$, and a close language $l^c$ compared to English. The distance between languages is calculated by the lang2vec (Littell et al., 2017), a tool that extracts features of different languages by querying the URIEL typological database[2]. Based on the relationship between language distance and transfer performance in (Ahmad et al., 2019), the threshold for language classification is set as 0.52. A language with a distance of more than 0.52 from English is a distant language, otherwise, it is a close language. Next, we introduce the two blocks and losses.

### 4.3.1 Syntactic Encoder Block

Many studies have found that PLMs can encode syntactic structures of sentences (Hewitt and Manning (2019); Chi et al. (2020)). For learning syntactic representation, we design a syntactic encoder block consisting of the multilingual PLM layer and the syntactic linear layer. Specifically, for an input text $x = \{w_i\}_{i=1}^{a}$, the output representations $\mathrm{h}(x) \in \mathbb{R}^{a \times b}$ from the *frozen* multilingual PLM are fed into the syntactic linear layer (a matrix $B \in \mathbb{R}^{b \times c}$). Then $\mathrm{h}(x)$ is transformed into universal syntactic space $g(x) = B\mathrm{h}(x)$, so the syntactic vectors of $i$th word $w_i$ can be defined as:

$$\boldsymbol{s}_i = B\mathrm{h}(w_i) \quad (5)$$

Inspired by Hewitt and Manning (2019), we adopt the syntactic labels from Universal Dependencies (UD[3]) to learn syntactic embedding with two tasks: depth prediction of a word and distance prediction of two words in the parse tree $T$. The losses of these two subtasks are defined as:

$$L_{depth} = \sum_i (|w_i| - \|\boldsymbol{s}_i\|_2^2), \quad (6)$$

$$L_{distance} = \sum_{i,j} \left| \mathrm{d}_T(w_i, w_j) - \mathrm{d}_B(\boldsymbol{s}_i, \boldsymbol{s}_j) \right| \quad (7)$$

---

[2]http://www.cs.cmu.edu/ dmortens/projects/7_project
[3]https://universaldependencies.org

where $|w_i|$ is the parse depth of a word defined as the number of edges from the root of the parse tree to $w_i$, and $\|\boldsymbol{s}_i\|_2$ is the tree depth L2 norm of the syntactic vector. $\mathrm{d}_T(w_i, w_j)$ is the number of edges in the path between the $i$th and $j$th word in the parse tree. As for $\mathrm{d}_B(\boldsymbol{s}_i, \boldsymbol{s}_j)$, it can be defined as the squared $L_2$ distance:

$$\mathrm{d}_B(\boldsymbol{s}_i, \boldsymbol{s}_j) = (\boldsymbol{s}_i - \boldsymbol{s}_j)^T(\boldsymbol{s}_i - \boldsymbol{s}_j) \quad (8)$$

To induce parse trees, we minimize the summation of the above two losses $L_{depth}$ and $L_{distance}$ and define the syntactic tree loss (STL) as:

$$L_{STL} = L_{depth} + L_{distance} \quad (9)$$

### 4.3.2 Distance Metric Block

Syntactic information is at the word level, so we introduce Word Mover's Distance (WMD) (Kusner et al., 2015) to calculate text syntactic distance incorporating the dissimilarity between word pairs. WMD is the cost of transporting a set of word vectors to the other in an embedding space.

Formally, the three-way parallel texts $(x, x^d, x^c)$ can be encoded into a syntactic space by our syntactic encoder to obtain the vectors $(\boldsymbol{s}, \boldsymbol{s}^d, \boldsymbol{s}^c)$. The inputs of WMD are probability weight and transportation cost function. As shown in the upper of Figure 2(b), the transportation cost function is:

$$\boldsymbol{c}^d = 1 - cos(\boldsymbol{s}, \boldsymbol{s}^d), \boldsymbol{c}^c = 1 - cos(\boldsymbol{s}, \boldsymbol{s}^c) \quad (10)$$

where $cos(\cdot, \cdot)$ is cosine similarity function. Following (Yokoi et al., 2020), we use *norm* of a word vector as the probability weight, and the norm of $i$th word vector $\boldsymbol{s}_i$ is:

$$a_i = \|\boldsymbol{s}_i\| \quad (11)$$

Therefore, the syntactic distance is defined as:

$$d_W^d = \sum_{i,j=1}^{N} \mathrm{T}_{ij}c^d, d_W^c = \sum_{i,j=1}^{N} \mathrm{T}_{ij}c^c, \quad (12)$$

where $\mathrm{T}$ is a transportation matrix, which is the solution for WMD to get the minimum cumulative cost of moving $\boldsymbol{s}$ to target. The transport values $\mathrm{T}_{ij}$ or $\mathrm{T}_{ji}$ is subjected to the probability weight $a_i$ or $a_j$, respectively. In the experiment, we use EMD (Yu and Herman, 2005) class of the python package cv[4] to solve the T. The triplet loss is:

$$L_{dis} = max\{0, m - d_W^d + d_W^c\}, \quad (13)$$

---
[4]https://pypi.org/project/cv/

where the $m$ represents a margin between a distant language and a close language from English.

Finally, the loss of the Syntactic Distance Metric Model (SDMM) is:

$$L = L_{STL} + L_{STL}^d + L_{STL}^c + L_{dis}, \quad (14)$$

where $(L_{STL}, L_{STL}^d, L_{STL}^c)$ are syntactic tree losses of three-way parallel texts.

### 4.4 Other Data Selection Strategies

In this section, we introduce other two data selection strategies to instantiate the meta-task collector. **Semantic Similarity Sampling** was used in recent works (Wu et al., 2020a; M'hamdi et al., 2021) to construct meta-tasks. In the experiment, we follow (M'hamdi et al., 2021) to use the cross-lingual extension to SBERT's pre-trained model (Reimers and Gurevych, 2019, 2020) as encoder block and use cosine similarity algorithm in distance metric block. For NER and MRC, the input is only the text of the task and the concatenation of the question and paragraph, respectively.
**Task-level Similarity Sampling** is proposed for tasks with long input text or multiple input texts because semantic similarity sampling ignores task information between texts. We use a pre-trained model fine-tuned on the English dataset as an encoder, called a task-specific pre-trained model. The distance metric block uses the cosine similarity algorithm. For MRC or NER, the input is the form of task fine-tuning followed by Devlin et al. (2019).

## 5 Experiments

### 5.1 Languages and Datasets

We evaluate the performance of our framework on NER and MRC benchmarks from XTREME (Hu et al., 2020) in 13 target languages, including Afrikaans (af), Arabic (ar), Bengali (bn), Finnish (fi), Javanese (jv), Indonesian (id), Korean (ko), Russian (ru), Swahili (sw), Telugu (te), Tagalog (tl), Yoruba (yo), and Chinese (zh). Among them, two languages (bn and te) are considered low-resource, while five languages (sw, af, tl, jv, and yo) are classified as extremely low-resource according to the classification method described in Bang et al. (2023). Additional details on language classification and statistics can be found in Appendix A.

For MRC, we use the gold passage version of the TydiQA dataset (TydiQA-GoldP) (Clark et al., 2020). It is more challenging than XQuAD (Artetxe et al., 2020) and MLQA (Lewis et al.,

| Model | TydiQA-GoldP (EM) | | | | | | | | |
|---|---|---|---|---|---|---|---|---|---|
| | ru | ar | fi | id | ko | bn | te | sw | avg |
| **mBERT** | | | | | | | | | |
| PRE† (Hu et al., 2020) | 38.8 | 42.8 | 45.3 | 45.8 | 50.0 | 32.7 | 38.4 | 37.9 | 41.5 |
| PRE | 39.4 | 44.8 | 43.2 | 48.5 | 46.6 | 35.2 | 38.4 | 39.9 | 42.0 |
| X-METRA (M'hamdi et al., 2021) | 48.9±0.4 | 63.3±0.8 | 59.1±1.1 | 65.2±0.5 | – | 39.0±1.9 | 49.7±0.5 | 61.4±0.4 | 55.2 |
| FT | 52.0±0.5 | 62.1±0.4 | 59.4±1.5 | 65.4±1.5 | 48.8±1.0 | 49.7±0.9 | 61.1±3.2 | 57.6±0.6 | 57.0 |
| FT w/syn_sample | 51.7±0.7 | 62.6±1.1 | 59.8±0.4 | 64.1±0.5 | 49.5±1.2 | 49.4±1.6 | 62.9±0.4 | 60.1±1.5 | 57.5 |
| XMT$_{random}$ | 52.8±0.9 | 62.2±0.3 | 61.5±0.9 | 66.4±1.4 | 51.3±3.7 | 49.9±1.5 | 62.5±0.9 | 62.1±0.5 | 58.6 |
| **Ours** | | | | | | | | | |
| MeTaCo-XMT$_{sem}$ | 51.7±0.3 | 62.7±0.3 | 61.5±0.9 | 65.7±0.8 | 50.9±0.2 | 48.1±2.4 | 62.7±0.3 | **64.3**±1.5 | 58.4 |
| MeTaCo-XMT$_{task}$ | 52.7±0.6 | **63.5**±0.2 | 61.2±0.7 | 66.0±0.9 | 51.3±1.3 | 52.8±1.2 | 64.0±0.3 | 63.4±0.3 | 59.4 |
| MeTaCo-XMT$_{syn}$ | **53.0**±0.5 | 63.4±0.3 | **61.6**±0.7 | **66.8**±0.4 | **51.9**±0.4 | **54.3**±0.9 | **64.5**±0.2 | 64.3±0.5 | **60.0** |
| **XLM-R$_{large}$** | | | | | | | | | |
| PRE† (Hu et al., 2020) | 42.1 | 40.4 | 53.2 | 61.9 | 10.9 | 47.8 | 43.6 | 48.1 | 45.0 |
| PRE | 41.9 | 55.2 | 56.5 | 64.2 | 47.4 | 50.1 | 54.7 | 52.7 | 53.7 |
| FT | 54.2±0.9 | 63.2±0.7 | 64.7±1.1 | 71.8±1.0 | 51.5±2.6 | 60.9±2.8 | 52.1±1.6 | 68.5±2.1 | 60.6 |
| FT w/syn_sample | 53.6±0.5 | 62.9±1.0 | 66.0±1.1 | 70.7±1.0 | 52.4±1.8 | 60.4±2.7 | 65.3±0.8 | 66.7±1.2 | 62.3 |
| XMT$_{random}$ | 55.1±0.8 | 64.5±1.6 | 65.2±0.8 | 72.6±0.5 | 55.8±0.7 | 66.7±3.0 | 67.4±0.8 | 69.9±0.2 | 64.7 |
| **Ours** | | | | | | | | | |
| MeTaCo-XMT$_{sem}$ | 55.7±0.9 | 64.8±1.4 | 65.3±0.6 | 71.0±0.4 | 53.9±0.5 | 66.1±0.6 | 66.5±0.4 | **71.7**±0.2 | 64.4 |
| MeTaCo-XMT$_{task}$ | **56.0**±1.0 | 64.6±1.2 | 65.6±0.9 | **73.0**±0.3 | 55.1±0.7 | 67.3±0.9 | 67.4±0.9 | **71.7**±1.3 | 65.1 |
| MeTaCo-XMT$_{syn}$ | **56.0**±0.3 | **65.3**±1.1 | **66.3**±0.4 | 72.9±0.4 | **56.9**±0.5 | **68.4**±0.7 | **67.5**±0.1 | 71.6±0.4 | **65.6** |

Table 1: EM score and standard deviation of 8 target languages and average on the TydiQA-GoldP dataset.

2020) as questions have been written without seeing the answers. The dataset is segmented following M'hamdi et al. (2021), with English training data as *Train* and 10% of training data from other languages as *Dev* for few-shot or meta-transfer. The provided test sets are used for evaluation. For the NER task, we employ the multilingual WikiAnn dataset (Pan et al., 2017), reserving $num = 100$ instances from the training data of other languages as *Dev*. Appendix B.1 provides detailed dataset statistics.

## 5.2 Baselines

mBERT (Pires et al., 2019) and XLM-R$_{large}$ (Conneau et al., 2020) are used as the base PLM. We compared our model with the following baselines:

- *PRE*: An initial task-specific base PLM baseline is fine-tuned on the English *Train* and evaluated on other languages *Test*.
- *FT*: A standard few-shot transfer baseline to fine-tune the *PRE* on target language *Dev*.
- *FT w/syn_sample*: We fine-tune the *PRE* model on *Dev* split of the target languages and the selected English support dataset by syntactic sampling method in Section 4.3.
- *XMT$_{randome}$*: The framework is similar to Figure 2(a), except the data selection is implemented by random sampling.

For TydiQA-GoldP, we focused on the baseline *X-METRA* (M'hamdi et al., 2021) which shares a similar setting with our framework, utilizing semantic similarity sampling based on (paragraph,

question, answer) triples. For generality, we only concatenate the paragraph and question.

For WikiAnn, we added the competitive and challenging zero-shot baselines with pseudo-labeled data, including *CROP*(Yang et al., 2022) and *SL_LEU*(Xu et al., 2021). They leverage translation or self-training methods to obtain pseudo-labeled data for target languages. Further details on these baselines can be found in Appendix C.

## 5.3 Implementation Details

MeTaCo-XMT is initialized by *PRE* following the hyper-parameters settings in XTREME (Hu et al., 2020). The meta-task collector employs English *Train* data as support sets and target language *Dev* data as query sets. We implemented the MAML using the *learn2learn*[5] library. *MeTaCo-XMT$_{syn}$*, *MeTaCo-XMT$_{task}$* and *MeTaCo-XMT$_{sem}$* respectively represent the framework of meta-tasks constructed by syntactic similarity sampling, task-similarity sampling, and semantic similarity sampling. For each model (except that *PRE* uses a fixed seed 42), we run 3 random initialization and report the average and standard deviation.

For SDMM, we collected a syntax-labeled corpus of 7k instances from the UD 2.7 Treebank (Zeman et al., 2020), covering 7 distant languages and 7 languages close to English (detailed information can be found in Appendix B.2). We utilize Universal HEAD tags in UD 2.7 for optimizing the syntactic tree loss. Further hyper-parameter details can be found in Appendix D.

---

[5]https://www.cnpython.com/pypi/learn2learn

| | **WikiAnn** (F1) | | | | | | | | | | | | | |
|---|---|---|---|---|---|---|---|---|---|---|---|---|---|---|
| **Model** | ru | zh | ar | fi | id | ko | bn | te | af | jv | sw | tl | yo | avg |
| mBERT | | | | | | | | | | | | | | |
| PRE†(Hu et al., 2020) | 64.0 | 42.7 | 41.1 | 77.2 | 53.5 | 59.6 | 70.0 | 48.5 | 78.9 | 62.5 | 67.5 | 73.2 | 33.6 | 59.4 |
| PRE | 61.9 | 43.3 | 46.2 | 76.8 | 58.5 | 59.9 | 67.6 | 49.2 | 75.5 | 56.8 | 68.6 | 68.4 | 51.1 | 60.3 |
| CROP (Yang et al., 2022) | 69.7 | 54.4 | 48.0 | 79.1 | 46.4 | 62.6 | 74.9 | 61.6 | 81.0 | 57.7 | 68.3 | 75.5 | 52.6 | 64.0 |
| SL-LEU (Xu et al., 2021) | **79.9** | 54.8 | 70.0 | **86.2** | 53.4 | **71.8** | **83.6** | 69.9 | 81.5 | 65.3 | 70.4 | 81.3 | 43.5 | 70.1 |
| FT | 75.8±0.6 | 56.6±0.8 | 72.4±0.3 | 81.0±0.1 | 83.4±3.2 | 66.7±0.6 | 70.9±1.2 | 67.9±0.4 | 80.9±0.4 | 83.8±0.3 | 83.1±1.6 | 78.3±0.7 | 81.8±1.0 | 75.6 |
| FT w/syn_sample | 75.7±1.0 | 59.7±0.4 | 73.5±0.3 | 79.8±1.0 | 81.9±0.9 | 70.5±0.3 | 72.0±1.9 | 65.7±0.1 | 82.4±0.5 | 76.7±1.1 | 85.5±0.4 | 81.3±0.7 | 90.7±1.3 | 76.6 |
| XMT$_{random}$ | 75.3±0.5 | 59.4±0.7 | 74.3±0.8 | 82.0±0.3 | 83.8±0.4 | 70.0±0.6 | 74.2±1.1 | 73.8±0.7 | 81.4±0.3 | 80.4±0.9 | 85.7±0.1 | 76.3±1.7 | 91.9±1.1 | 77.6 |
| **Ours** | | | | | | | | | | | | | | |
| MeTaCo-XMT$_{sem}$ | 75.2±0.5 | 60.0±0.6 | **75.1**±0.4 | 82.0±0.2 | 84.2±0.1 | 69.8±0.1 | 74.2±2.2 | 73.1±0.6 | 82.0±0.1 | 81.4±1.1 | 86.3±0.8 | 78.2±0.4 | 92.3±0.5 | 78.0 |
| MeTaCo-XMT$_{task}$ | 75.6±0.3 | 59.8±0.2 | 74.8±0.6 | 82.2±0.3 | 84.1±0.1 | 70.1±0.1 | 73.2±1.0 | 73.1±0.6 | 81.8±0.2 | 91.8±0.2 | 85.9±1.1 | 77.1±1.0 | 91.8±0.2 | 78.6 |
| MeTaCo-XMT$_{syn}$ | 77.6±0.2 | **61.9**±0.4 | 74.7±0.1 | 82.2±0.1 | **84.6**±0.3 | 71.7±0.0 | 76.2±0.4 | **74.6**±0.8 | **84.5**±0.4 | **93.7**±0.2 | **88.0**±0.3 | **81.8**±0.6 | **93.8**±0.4 | **80.4** |
| XLM-R$_{large}$ | | | | | | | | | | | | | | |
| PRE†(Hu et al., 2020) | 69.1 | 33.1 | 53.0 | 79.2 | 53.0 | 60.0 | 78.8 | 55.8 | 78.9 | 62.5 | 70.5 | 73.2 | 33.6 | 61.6 |
| PRE | 71.6 | 26.5 | 57.6 | 81.5 | 55.2 | 63.2 | 78.0 | 59.6 | 77.3 | 63.0 | 68.4 | 74.1 | | 62.8 |
| FT | 80.2±1.9 | 48.3±7.5 | 77.0±2.4 | **85.2**±1.1 | 78.0±5.6 | 72.2±3.2 | 77.3±4.9 | 70.1±6.2 | 83.0±0.4 | 85.4±1.1 | 86.7±2.4 | 78.6±1.3 | 84.2±2.4 | 77.4 |
| FT w/syn_sample | 79.8±0.3 | 49.9±5.0 | 76.1±2.4 | 82.7±0.2 | 86.2±0.5 | 70.6±2.4 | 77.2±2.0 | 71.4±2.2 | 82.4±0.5 | 84.9±1.2 | 86.7±0.6 | 76.9±0.9 | 89.0±2.2 | 77.9 |
| XMT$_{random}$ | 80.7±0.5 | 53.9±1.6 | 77.7±0.7 | 83.8±0.5 | 82.9±4.6 | 72.4±0.6 | 79.8±0.9 | **77.5**±0.6 | **84.0**±0.1 | 80.2±5.3 | 86.8±0.8 | **80.5**±0.7 | 86.9±4.4 | 79.0 |
| **Ours** | | | | | | | | | | | | | | |
| MeTaCo-XMT$_{sem}$ | 80.7±0.6 | 51.4±4.6 | 76.0±0.6 | 83.2±0.1 | 85.4±0.0 | 74.8±1.3 | 80.1±1.4 | 74.1±0.9 | 83.1±0.4 | 79.2±0.3 | 87.8±0.3 | 77.1±1.1 | 91.8±1.3 | 78.8 |
| MeTaCo-XMT$_{task}$ | 79.7±1.6 | 53.3±0.3 | 77.1±0.1 | 82.9±0.1 | 85.6±0.1 | 73.7±1.3 | 80.7±0.2 | 75.2±0.5 | 82.9±0.8 | 81.2±0.9 | 87.1±0.1 | 78.0±0.4 | 89.1±0.8 | 79.0 |
| MeTaCo-XMT$_{syn}$ | **80.9**±0.3 | 54.3±0.2 | **78.1**±0.1 | 83.5±0.2 | **87.8**±0.2 | 74.9±0.1 | 82.1±0.6 | 75.7±0.2 | 83.8±0.1 | 85.8±0.3 | 87.2±0.5 | 77.9±0.3 | 92.1±0.2 | 80.3 |

Table 2: F1 score and standard deviation of 13 languages and average on the WikiAnn dataset.

| | **Languages** | | | | | | | | | | | | | |
|---|---|---|---|---|---|---|---|---|---|---|---|---|---|---|
| **Model** | ru | zh | ar | fi | id | ko | bn | te | af | jv | sw | tl | yo | avg |
| *TydiQA-GoldP* (EM) | | | | | | | | | | | | | | |
| mBERT — MeTaCo-XMT$_{syn}$ | 53.0±0.5 | – | 63.4±0.3 | 61.6±0.7 | 66.8±0.4 | 51.9±0.4 | 54.3±0.9 | 64.5±0.2 | – | – | 64.3±0.5 | – | – | **60.0** |
| wo STL | 53.0±0.6 | – | 63.3±1.3 | 61.0±1.5 | 65.4±0.4 | 51.2±1.2 | 53.7±3.2 | 64.1±0.8 | – | – | 62.9±0.8 | – | – | 59.4 |
| WMD→cos. | **53.3**±1.3 | – | 63.0±0.3 | 60.8±0.3 | 65.8±0.9 | 51.8±0.5 | 51.6±2.9 | 64.2±0.5 | – | – | 62.1±0.6 | – | – | 59.1 |
| *WikiAnn* (F1) | | | | | | | | | | | | | | |
| mBERT — MeTaCo-XMT$_{syn}$ | **77.6**±0.2 | **61.9**±0.4 | **74.7**±0.1 | 82.2±0.1 | **84.6**±0.3 | **71.7**±0.0 | **76.2**±0.4 | **74.6**±0.8 | **84.5**±0.4 | **93.7**±0.2 | **88.0**±0.3 | 81.8±0.6 | **93.8**±0.4 | **80.4** |
| wo STL | 75.1±1.1 | 59.9±0.4 | 73.9±0.4 | **82.4**±0.4 | 83.9±0.3 | 69.5±0.6 | 74.1±0.6 | 74.0±0.5 | 84.1±0.6 | 92.1±0.4 | 85.4±1.1 | **82.3**±1.3 | 88.6±1.1 | 78.9 |
| WMD→cos. | 75.0±1.2 | 60.0±0.4 | 73.8±0.2 | **82.4**±0.4 | 83.8±0.2 | 69.8±0.6 | 73.6±0.8 | 73.9±0.5 | 84.0±0.5 | 91.7±0.4 | 85.5±1.0 | 82.1±1.4 | 88.6±1.1 | 78.8 |

Table 3: Ablation results for MRC and NER task based on mBERT.

## 5.4 Results

Table 1 shows the results of the Exact Match (EM) score on TydiQA-GoldP, while F1 scores are reported in Table 9 of Appendix E.1. Our method based on SDMM (i.e. MeTaCo-XMT$_{syn}$) is superior to the baselines in terms of EM for all 8 target languages. Notably, MeTaCo-XMT$_{syn}$ achieves a significant EM improvement of 4.4%, 1.6%, and 2.2% over the mBERT baseline for the low-resource languages Bengali (bn), Telugu (te), and the extremely-low resource language Swahili (sw), respectively. MeTaCo-XMT$_{syn}$ based on mBERT and XLM-R$_{large}$ demonstrates average EM improvements of 1.4% and 0.9% compared to the strong baseline XMT$_{random}$. Moreover, among the models with reported standard deviations, MeTaCo-XMT$_{syn}$ exhibits the highest stability, indicated by a lower average standard deviation.

The results of WikiAnn in 13 languages are shown in Table 2. The MeTaCo-XMT$_{syn}$ model achieves an average improvement of 2.8% and 1.3% over the strong baseline XMT$_{random}$ on mBERT and XLM-R$_{large}$, respectively. Our mBERT-based MeTaCo-XMT outperforms other models on all 5 extremely-low resource languages and exhibits the lowest average standard deviation. These findings validate the effectiveness of syntac-

tic similarity sampling for structural tasks like NER. Overall, the results from both tasks demonstrate the efficacy of our MeTaCo-XMT framework.

**The results of three data selection strategies** and random sampling strategy show that MeTaCo-XMT$_{syn}$ obtains best performance and with lower standard deviation. Semantic similarity sampling is less effective than random sampling may due to limited diversity in the support samples caused by high semantic similarity among query samples. The task-similarity sampling strategy is better than the semantic similarity sampling, demonstrating that task information is significant for meta-task selection in MRC and NER.

**Ablation study** was conducted based on the MeTaCo-XMT$_{syn}$ model with different structures of SDMM. We study the effectiveness of the syntactic tree loss (STL) and WMD metric, as shown in Table 3. The results show the importance of all proposed components, as removing syntactic tree loss (STL) or using cosine similarity instead of Word Mover's Distance (WMD) leads to a slight decrease in average effectiveness across both tasks.

## 6 Analysis

Our analytical experiments consider the performance of the four cross-lingual meta-transfer models with random, semantic-similar, task-similar,

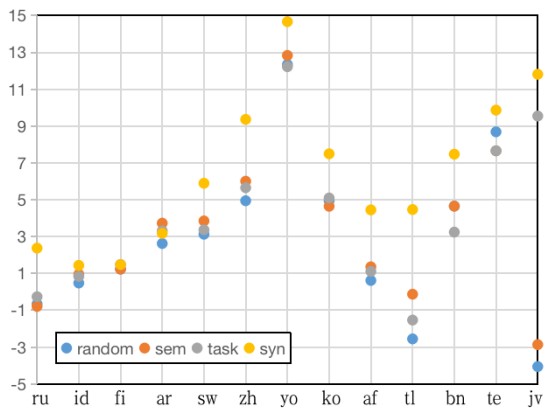

Figure 3: The effect gain $\delta$ relative to the *FT* model in 13 language of WikiAnn. The distances between the language and English increase from left to right. Languages starting with Arabic (ar) are distant languages.

and syntactic-similar samples named 'random', 'sem', 'task', and 'syn' for short, respectively.

**Syntactic Distance Analysis** Our MeTaCo-XMT framework significantly improved the average performance on NER task, especially MeTaCo-XMT$_{syn}$. Therefore, we explored the influence of different sampling strategies on languages with different syntactic distances from English (calculated by lang2vec (Littell et al., 2017)). As shown in Figure 3, the y-axis is the effect gain percent $\delta$ relative to *FT* model based on mBERT:

$$\delta = \frac{F1_{\text{MeTaCo-XMT}} - F1_{\text{FT}}}{F1_{\text{FT}}} \quad (15)$$

The results show that MeTaCo-XMT$_{syn}$ always obtains the positive gain in 13 languages, and the syntactic similarity sampling schema is more advantageous in distant languages from English than other strategies.

**Case Study for Meta-task** In order to observe the sampling effect of different data selection strategies, an example of English (en)-Indonesian (id) match pairs in WikiAnn are listed in Table 4 (more cases shown in Appendix E.5). In example #1, original Indonesian and *syn* English sentences have similar syntax and even semantics, and the *task* sentence has a similar syntax to the original sentence. *random* and *sem* sentences are no obvious connection to the Indonesian query sentence. Combining the results of four strategies on NER, the structurally similar examples might benefit cross-lingual meta-transfer learning. Furthermore, the structurally similar examples can more effectively stimulate the ability of meta-learning in cross-lingual

| #1 | id: Lagu ini ditulis oleh [Matthew Bellamy]$_{PER}$ .
en: This song was written by [Matthew Bellamy]$_{PER}$. |
|---|---|
| random | [The History of England from the Accession of James II]$_{ORG}$ " |
| sem | [Thom Bell]$_{PER}$ – composer |
| task | The artwork was credited to [Arnold Roth]$_{PER}$. |
| syn | All tracks are written by [Jason Lytle]$_{PER}$. |

Table 4: A example of different data select strategies in WikiAnn.

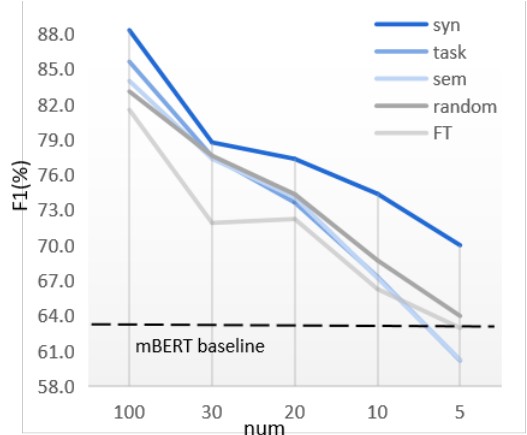

Figure 4: Avarage F1 in five extremely-low resource languages of NER task with different $num$.

transfer, as analyzed in Appendix E.2.

**Query Data Size Analysis** In the few-shot cross-lingual transfer scenario, the target language has a certain number of annotated data. Low resource languages are challenging to collect annotation data, so we explore (i) whether a small amount of data can achieve an improvement and (ii) the robust performance of different data selection strategies with different sizes $num$ of query instances.

With different $num$ settings, we reported the average F1 in five extremely-low resource languages of the NER task (the entire results can be found in Appendix E.4). As shown in Figure 4, even with only 5 query samples, the method MeTaCo-XMT$_{syn}$ based on syntactic similarity sampling can achieve the best results with great improvement than strong baseline *FT* and XMT$_{random}$. For MeTaCo-XMT$_{task}$ and MeTaCo-XMT$_{sem}$, they are not suitable for only 5 query instances; and when the query examples are less than 30, the effect is lower than XMT$_{random}$. So *task* and *sem* strategies are suitable for scenarios with a certain amount of target language data. However, even when query data is scarce, XMT$_{random}$ always has an advantage. This may be because randomly sampled instances preserve the diversity of the samples.

According to the better performance of MeTaCo-XMT$_{syn}$ and the cases study, the sample selection with syntactic similar conforms to the syntactic distribution of the target language and ensures the diversity of samples.

## 7 Conclusion

In this paper, we have presented a novel meta-task collector-based cross-lingual meta-transfer framework, which can adapt different meta-task selection strategies to construct meta-transfer training data, reducing the cross-lingual performance gap between languages due to language differences. To close the syntactic distance between languages, we propose the syntactic distance metric model that encodes text pairs to syntactic space and selects meta-task by WMD for meta-transfer learning. Two other data selection strategies are explored: semantic similarity sampling and task-similarity sampling. We demonstrate the validity of our framework on both the NER and MRC tasks, especially the syntactic similar sampling-based method reaches a new state-of-the-art for most languages. Further analyses suggest that the proposed MeTaCo-XMT with syntactic similar sampling can effectively improve the cross-lingual transfer performance with only a small amount of data, especially for low-resource languages.

## Acknowledgement

This work is supported by the Key Research and Development Program of Zhejiang Province (No. 2021C01013), the Fundamental Research Funds for the Central Universities (No. 226-2023-00060), National Key Research and Development Project of China (No. 2018AAA0101900), Alibaba-Zhejiang University Joint Research Institute of Frontier Technologies, and MOE Engineering Research Center of Digital Library.

## 8 Limitations

Due to the computation constraints, we were not able to experiment with our framework on more NLP tasks with more languages, which will be supplemented in the future. The syntactic difference is a key factor affecting cross-lingual, and syntactic distance can help to understand or quantify the transfer differences. Our framework with SDMM is a simple attempt to use syntactic distance metrics to construct meta-task and reduce the language gap in cross-lingual transfer learning.More work needs to be done on syntactic differences and syntactic distance metrics.

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

# Appendices

## A  Classify the resource levels for Languages

Following Bang et al. (2023) and Lai et al. (2023), the 13 languages in our study are grouped into categories based on their data ratios in the CommomCrawl corpus[6](i.e., the main data to pre-train multilingual language models). In particular, a language will be considered as High Resource (H), Medium Resource (M), Low Resource (L), and Extremely-Low Resource (X), if its data ratio is greater than 1% (> 1%), between 0.1% and 1% (> 0.1%), between 0.01% and 0.1% (> 0.01%), and smaller than 0.01% (< 0.01%) respectively. Table 5 presents information and categories for the languages considered in our work.

## B  Dataset Statistics

### B.1  Benchmark Dataset

Tables 6 and 7 show the statistics of WikiAnn and TydiQA-GoldP respectively per language and split in our study.

---

[6]https://commoncrawl.github.io/cc-crawl-statistics/plots/languages.html

| Language | Code | Pop. (M) | CC size (%) | Cat. |
|---|---|---|---|---|
| English | en | 1,452 | 43.8846 | H |
| Russian | ru | 258 | 9.2012 | H |
| Chinese | zh | 1.118 | 5.1984 | H |
| Indonesian | id | 199 | 0.7399 | M |
| Arabic | ar | 274 | 0.6688 | M |
| Korean | ko | 81 | 0.5944 | M |
| Finnish | fi | 4.9 | 0.3535 | M |
| Bengali | bn | 272.7 | 0.0454 | L |
| Telugu | te | 95.7 | 0.017 | L |
| Swahili | sw | 71 | 0.0074 | X |
| Afrikaans | af | 6.2 | 0.0072 | X |
| Tagalog | tl | 72 | 0.0068 | X |
| Javanese | jv | 60 | 0.0012 | X |
| Yoruba | yo | 42 | 0.0004 | X |

Table 5: List of languages, language codes, numbers of first and second speakers, data ratios in the Common-Crawl corpus, and language categories.

| Lang | ISO | Train | Dev | Test |
|---|---|---|---|---|
| English | en | 20,000 | – | – |
| Afrikaans | af | 5,000 | 100 | 1,000 |
| Arabic | ar | 20,000 | 100 | 10,000 |
| Bengali | bn | 10,000 | 100 | 1,000 |
| Finnish | fi | 20,000 | 100 | 10,000 |
| Javanese | jv | 100 | 100 | 100 |
| Indonesian | id | 20,000 | 100 | 10,000 |
| Korean | ko | 20,000 | 100 | 10,000 |
| Russian | ru | 20,000 | 100 | 10,000 |
| Swahili | sw | 1,000 | 100 | 1,000 |
| Telugu | te | 1,000 | 100 | 1,000 |
| Tagalog | tl | 10,000 | 100 | 1,000 |
| Yoruba | yo | 100 | 100 | 100 |
| Chinese | zh | 20,000 | 100 | 10,000 |

Table 6: 13 languages statistics of WikiAnn dataset in our study.

| Lang | ISO | Train | Dev | Test |
|---|---|---|---|---|
| English | en | 3,326 | – | – |
| Arabic | ar | 13,324 | 1,481 | 921 |
| Bengali | bn | 2,151 | 239 | 113 |
| Finnish | fi | 6,169 | 686 | 782 |
| Indonesian | id | 5,131 | 571 | 565 |
| Korean | ko | 1462 | 163 | 276 |
| Russian | ru | 5,841 | 649 | 812 |
| Swahili | sw | 2,479 | 276 | 499 |
| Telugu | te | 5,006 | 557 | 669 |

Table 7: Statistics of TydiQA-GoldP dataset per language and split.

| Lang | Script | Family | Dist. to English | Cat. |
|---|---|---|---|---|
| English | Latin | Indo-European | – | – |
| Spanish | Latin | Indo-European | 0.4 | Clos. |
| German | Latin | Indo-European | 0.42 | Clos. |
| French | Latin | Indo-European | 0.46 | Clos. |
| Icelandic | Latin | Indo-European | 0.47 | Clos. |
| Portuguese | Latin | Indo-European | 0.47 | Clos. |
| Russian | Cyrillic | Indo-European | 0.49 | Clos. |
| Italian | Latin | Indo-European | 0.51 | Clos. |
| Thai | Thai | Tai-Kadai | 0.56 | Dist. |
| Chinese | Han (Traditional) | Sino-Tibetan | 0.57 | Dist. |
| Arabic | Arabic | Afro-Asiatic | 0.57 | Dist. |
| Hindi | Devanagari | Indo-European | 0.59 | Dist. |
| Korean | Hangul | Koreanic | 0.62 | Dist. |
| Japanese | Japanese | Japonic | 0.66 | Dist. |
| Turkish | Latin | Turkic | 0.7 | Dist. |

Table 8: List of Languages, languages script, language family, and the distance to English. A distance larger than 0.53 is the distant language (Dist.) to English, and conversely is the close language (Clos.) to English.

### B.2 Dataset for SDMM

The training data of SDMM is from UD 2.7 Treebank. In particular, we select 15 languages (including English) from Parallel Universal Dependencies (PUD) treebanks[7], each language containing 1000 sentences. According to the distance between each language and English, we construct a three-way parallel corpus with 7 distant languages and 7 close languages for SDMM, including 6500 data for training and 500 data for development. The distance information between each language and English is shown in Table 8.

### C  Baselines for NER

For WikiAnn, we compared our method with two high-performance zero-shot baselines:

- *CROP* (Yang et al., 2022): A Cross-lingual Entity Projection framework (CROP) with a multilingual labeled sequence translation model. It obtains the labels on the English NER model by translating the target language raw corpus (more than 100k instances) into English and then utilizes the multilingual labeled sequence translation model to obtain the labels of the target language corpus. The whole pipeline is integrated into an end-to-end NER model by way of self-training.

- *SL-LEU* (Xu et al., 2021): A self-learning framework (SL) that further utilizes unlabeled data of target languages based on a model fine-tuned on English training data, combined with uncertainty estimation in the process to

---

[7]http://universaldependencies.org/conll17/

select high-quality silver labels. The best performance of SL for the NER task is achieved by adopting Language Heteroscedastic Uncertainty (LEU) as the uncertainty estimation. It uses all of the dev set of target languages on task data as the source of unlabeled data.

They leverage translation or self-training methods to obtain pseudo-labeled data for target languages.

## D Hyperparameters

For the meta-task collector, we set $q = k = 6$ or $q = k = 8$ to construct a meta-task. The number of meta-tasks is $[300, 400, 500, 600]$. The learning rate (lr) $\alpha$ and $\beta$ are 3e-5 on the NER task. For TydiQA, $\alpha$ is 3e-5 and $\beta$ is 1e-5. The step size $r$ for the inner loop is 4 or 3. The batch size of the outer loop is 4 or 5. The three seeds for NER and MRC tasks are $[111, 222, 3333]$. We experimented using $[3, 4, 5, 6, 7, 8]$ epoch number for meta-transfer training, referring the setting in M'hamdi et al. (2021); Liu et al. (2021); Nooralahzadeh et al. (2020). However, the 5 or 8-epoch setting led to the best results in our experiments.

For hyper-parameters in SDMM, we use a batch size of 16 and 20 for the model based on mBERT and XLM-R large baselines respectively, Adam with the lr of 1e-3, and the epoch is 20. The margins $m$ were 0.6 or 0.5, and the syntactic vector dimensions were 32 and 64 for SDMM based on mBERT and XLM-R large baselines respectively.

## E More Result

### E.1 F1 Score of TydiQA-GoldP

Tables 9 show the F1 score of TydiQA-GoldP in 8 target languages. MeTaCo-XMT$_{syn}$ outperforms the strong baselines based on mBERT and XLM-R by 1.4% and 0.6% on average, respectively. And its standard deviation is also significantly lower than the random sampling-based method XMT$_{random}$. For extremely-low resource language, our MeTaCo-XMT framework can always outperform XMT$_{random}$ with a lower standard deviation.

### E.2 Data Selection in Meta-learning and Fine-tuning

We extended the baseline *FT w/syn_sample* to experiment on four data selection methods (called *FT w/sample*), and the results of the NER task (except rich-resource languages) are shown in Figure 5.

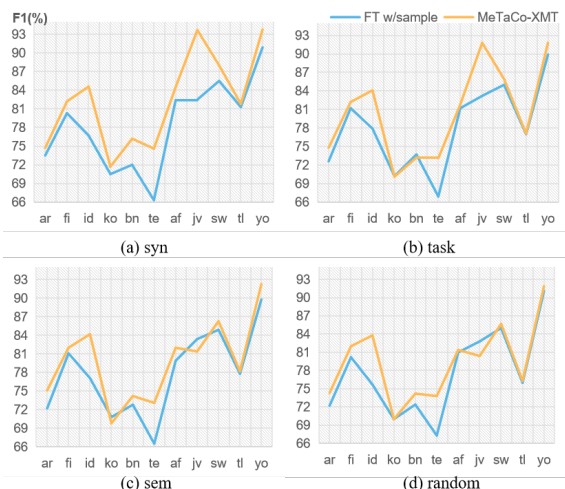

Figure 5: The results of MeTaCo-XMT and FT w/sample in NER task (except rich-resource languages) with four data selection methods.

The results of our MeTaCo-XMT framework are also shown in Figure 5 to compare the performance of different learning methods (meta-learning and fine-tuning) using these selected data.

Overall, meta-learning methods outperform fine-tuning methods in most languages, especially in medium-resource languages. From the comparison of different data selection strategies, MeTaCo-XMT$_{syn}$ always outperforms *FT w/syn_sample*. MeTaCo-XMT$_{task}$ can also surpass the fine-tuning method in most languages. However, MeTaCo-XMT$_{sem}$ and XMT$_{random}$ are difficult to outperform the *FT w/sample* in extremely-low resource languages. The examples based on syntactically similar selection can more effectively stimulate the ability of meta-learning in cross-lingual transfer learning.

### E.3 Comparison with Translation-train Baselines

The method fine-tuned on translation-train data is a strong baseline, such as translate-train baseline in XTREME (Hu et al., 2020) and translate-train setting in ByT5 (Xue et al., 2022). They fine-tuned on one or all target languages data translated from English data, which is significantly more than the DEV data (10% of training data in one target language) we used. Compared with ByT5 based on the mT5$_{base}$ model (580M parameter) in the translate-train setting, we can have improvement by 2.7 and 3.9 on the avarage of F1 and EM metric in TydiQA-GoldP. Furthermore, the results reported in the XTREME benchmark for the Translate-train base-

| Model | | TydiQA-GoldP (F1) | | | | | | | | |
|---|---|---|---|---|---|---|---|---|---|---|
| | | ru | ar | fi | id | ko | bn | te | sw | avg |
| mBERT | PRE†(Hu et al., 2020) | 60.0 | 62.2 | 59.7 | 64.8 | 58.8 | 49.3 | 49.6 | 57.5 | 57.7 |
| | PRE | 59.4 | 64.3 | 59.0 | 64.8 | 57.1 | 51.8 | 47.8 | 59.8 | 58.0 |
| | X-METRA (M'hamdi et al., 2021) | 66.1±0.1 | 78.4±0.6 | 72.7±0.4 | 77.7±0.2 | – | 53.2±0.5 | 66.6±0.4 | 71.7±0.2 | 69.5 |
| | FT | 68.3±0.7 | 77.5±0.4 | 72.0±1.0 | 77.9±1.0 | 59.0±0.7 | 63.4±1.0 | 76.5±2.6 | 69.2±0.5 | 70.5 |
| | FT w/syn_sample | 68.6±0.6 | 78.2±0.5 | 72.2±0.6 | 77.2±0.4 | 58.6±1.3 | 63.9±1.9 | 77.3±0.3 | 71.0±1.7 | 70.9 |
| | XMT$_{random}$ | 68.7±0.7 | 77.5±0.4 | **73.3**±0.9 | 77.8±1.1 | 60.4±3.2 | 62.1±2.2 | 76.6±0.9 | 71.6±1.1 | 71.0 |
| | **Ours** | | | | | | | | | |
| | MeTaCo-XMT$_{sem}$ | 67.4±0.2 | 77.8±1.3 | 72.7±0.7 | **78.2**±0.8 | 59.6±2.4 | 60.5±1.5 | 76.4±0.6 | 73.5±0.7 | 70.8 |
| | MeTaCo-XMT$_{task}$ | 68.3±0.4 | 78.9±0.5 | **73.3**±0.7 | 77.0±0.9 | 60.3±2.3 | 65.6±2.3 | 78.2±0.5 | 73.2±0.1 | 71.8 |
| | MeTaCo-XMT$_{syn}$ | **69.1**±0.7 | **79.0**±0.2 | 72.9±0.6 | 78.0±0.4 | **61.2**±0.7 | **67.0**±0.8 | **78.7**±0.3 | **73.9**±0.7 | **72.4** |
| XLM-R$_{large}$ | PRE†(Hu et al., 2020) | 67.0 | 67.6 | 70.5 | 77.4 | 31.9 | 64.0 | 70.1 | 66.1 | 64.3 |
| | PRE | 67.6 | 75.6 | 73.3 | 79.6 | 59.8 | 66.5 | 74.2 | 71.4 | 71.0 |
| | FT | 72.8 | 80.2 | 78.5 | 83.3 | 62.9 | 75.5 | 65.3 | 78.0 | 74.6 |
| | FT w/syn_sample | **75.0**±0.8 | 79.9±0.7 | **80.4**±0.9 | 83.6±0.8 | 64.1±1.6 | 76.2±1.0 | 80.8±0.6 | 77.0±0.7 | 77.1 |
| | XMT$_{random}$ | 73.5±0.9 | 80.5±0.9 | 78.4±1.2 | **83.7**±0.4 | **66.2**±1.4 | 79.2±2.4 | 82.1±0.7 | 78.4±0.3 | 77.7 |
| | **Ours** | | | | | | | | | |
| | MeTaCo-XMT$_{sem}$ | 73.4±0.1 | 80.7±1.0 | 78.6±0.3 | 83.4±0.7 | 64.6±0.8 | 76.8±1.4 | 81.5±0.2 | 79.7±0.3 | 77.3 |
| | MeTaCo-XMT$_{task}$ | 74.1±0.8 | **81.1**±0.6 | 78.8±1.2 | 83.3±0.2 | 64.9±0.7 | 79.9±2.8 | **82.5**±0.3 | 79.8±0.2 | 78.1 |
| | MeTaCo-XMT$_{syn}$ | 73.8±1.0 | **81.1**±0.5 | 79.2±0.9 | **83.7**±0.1 | 66.0±0.8 | **80.4**±2.1 | 82.4±0.5 | **80.2**±0.3 | **78.3** |

Table 9: F1 score and standard deviation of 8 target languages and average on the TydiQA-GoldP dataset.

| Model | | WikiAnn (F1) | | | | | |
|---|---|---|---|---|---|---|---|
| | | af | jv | sw | tl | yo | avg |
| mBert baseline | | 75.5 | 56.8 | 68.6 | 68.4 | 51.1 | 64.1 |
| $num=30$ | FT | 72.4±1.1 | 68.0±0.6 | 71.5±0.1 | 73.4±2.2 | 74.4±5.0 | 71.9 |
| | random | 80.0±0.4 | 76.1±0.2 | 83.2±0.4 | 75.8±0.4 | 73.2±1.2 | 77.7 |
| | sem | 78.5±0.2 | 75.4±1.1 | 83.1±0.5 | 74.7±1.1 | 75.3±1.4 | 77.4 |
| | task | 78.8±0.5 | **76.2**±0.1 | 83.0±0.4 | 74.7±1.1 | 75.3±1.4 | 77.6 |
| | syn | **82.0**±0.1 | 75.7±0.4 | **83.6**±0.1 | **77.0**±0.3 | **75.8**±0.5 | **78.8** |
| $num=20$ | FT | 77.5±0.2 | 62.5±2.2 | 71.8±0.5 | 74.1±3.0 | **75.4**±0.6 | 72.3 |
| | random | 79.5±0.3 | 72.3±0.7 | 82.3±0.5 | **76.5**±0.4 | 61.3±2.1 | 74.4 |
| | sem | 77.1±0.2 | 72.8±1.9 | 82.0±0.1 | 74.6±0.9 | 64.0±0.5 | 74.1 |
| | task | 77.1±0.2 | 72.7±1.8 | 82.0±0.1 | 74.7±1.0 | 61.9±2.1 | 73.7 |
| | syn | **81.2**±0.3 | **74.0**±0.4 | **83.6**±0.3 | 76.4±0.1 | 71.8±0.9 | **77.4** |
| $num=10$ | FT | 78.9±0.7 | 63.4±2.8 | 69.2±0.4 | 63.0±0.2 | 57.1±5.5 | 66.3 |
| | random | 77.2±0.2 | 61.9±0.3 | 79.4±0.4 | 64.7±1.2 | **60.6**±0.9 | 68.7 |
| | sem | 73.8±0.4 | 61.9±0.3 | 79.5±0.0 | 61.5±0.5 | 59.6±1.9 | 67.2 |
| | task | 73.8±0.4 | 62.1±0.5 | 79.5±0.1 | 61.5±0.5 | 59.9±1.6 | 67.4 |
| | syn | **80.3**±0.2 | **74.0**±0.4 | **83.6**±0.4 | **74.0**±0.2 | 60.2±0.6 | **74.4** |
| $num=5$ | FT | 75.9±1.2 | 62.8±0.5 | **68.1**±0.8 | 58.4±0.3 | 49.8±5.5 | 63.0 |
| | random | 74.7±0.3 | 57.0±0.6 | 67.2±0.4 | 61.4±0.6 | 59.7±1.2 | 64.0 |
| | sem | 68.2±0.3 | 57.1±0.9 | 62.5±1.2 | 54.5±0.3 | 59.1±0.4 | 60.3 |
| | task | 68.5±0.2 | 56.9±0.4 | 62.5±1.2 | 54.3±0.7 | 59.0±0.8 | 60.2 |
| | syn | **79.3**±0.2 | **65.9**±0.4 | 66.0±0.0 | **75.0**±0.3 | **63.9**±0.4 | **70.0** |

Table 10: F1 score and standard deviation of 5 extremely-low resource languages on the NER task with different $num$ setting.

line are also lower than our model in the TydiQA-GoldP task. Performance increasement demonstrates the advantages of our model in the few-shot scenario.

### E.4 The Results with Different Query Data Size

Table 10 shows the F1 score and standard deviation of 5 extremely-low resource languages on the NER task with different $num$ settings.

### E.5 Case Study

We report more data select cases in Table 11.

| | |
|---|---|
| **#2** | **fi:** Hän syntyi [Stralsundissa]$_{LOC}$ ja opiskeli [Leipzigin konservatoriossa]$_{ORG}$.
**en:** He was born in [Stralsund]$_{LOC}$ and studied at the [Leipzig Conservatory]$_{ORG}$. |
| random | 1997 : [Pust Mirom Pravit Lyubov]$_{ORG}$ ” |
| sem | The project went from [New Haven, Indiana to Toledo, Ohio]$_{LOC}$. |
| task | He moved to [Memphis , Tennessee]$_{LOC}$ with his family at the age of twelve. |
| syn | He was born in [Telečka]$_{LOC}$, [Zapadna Bačka]$_{ORG}$, [Serbia]$_{LOC}$. |
| **#3** | **id:** Saat ini ia bermain untuk [PSIS Semarang]$_{ORG}$.
**en:** He currently plays for [PSIS Semarang]$_{ORG}$. |
| random | [Crystal Tovar Aragón]$_{PER}$ |
| sem | He plays for [Thailand Premier League]$_{ORG}$ clubside [Samut Songkhram FC]$_{ORG}$. |
| task | [Regional District of Fraser-Cheam]$_{LOC}$ |
| syn | He currently plays for [Sivasspor]$_{ORG}$ in the [Super Lig]$_{ORG}$. |
| **#4** | **zh:** [蒋中正]$_{PER}$(中华民国总统、中国国民党总裁)
**en:** [Jiang Zhongzheng]$_{PER}$(President of the Republic of China, President of the Chinese Kuomintang) |
| random | His reign was also marked by the highly controversial execution of his son, [Prince Sado]$_{PER}$, in 1762 . |
| sem | ’ ” [Terengganu]$_{LOC}$ ” ’ |
| task | [Governor of Kentucky]$_{PER}$ : [William Owsley]$_{PER}$ ( [Whig ]$_{ORG}$) ( until September 6 ), [John J. Crittenden]$_{PER}$ ( [Whig]$_{ORG}$ ) ( starting September 6 ) |
| syn | [Bao Zheng]$_{PER}$ (包拯) |

Table 11: The examples of different data select strategies in the WikiAnn dataset of three target languages (Finnish(fi), Indonesian(id), and Chinese (zh)).