# OpenReview forum: "Good Meta-tasks Make A Better Cross-lingual Meta-transfer Learning for Low-resource Languages"
_EMNLP/2023/Conference — EMNLP 2023 Findings_

### Official Review · Reviewer_5wKC · 2023-08-03

**Soundness:** 4

**Excitement:**

3: Ambivalent: It has merits (e.g., it reports state-of-the-art results, the idea is nice), but there are key weaknesses (e.g., it describes incremental work), and it can significantly benefit from another round of revision. However, I won't object to accepting it if my co-reviewers champion it.

**Paper Topic And Main Contributions:**

This paper proposes to use a task collector to select better support tasks for the cross-lingual MAML algorithm. The collector is built based on a syntactic distance metric model, which is used to calculate the Word Mover’s Distance between the target task samples and the source task instances. The syntactic distance metric model contains a syntactic encoder block based on the pre-trained model and a distance metric block. Experiments show the superiority of the method over several baselines.

**Questions For The Authors:**

1.	L229-L230, why did you evaluate the optimized model on the target language test dataset directly? If my understanding is correct, the model should be optimized again after the meta-learning process to make the model better for the target task.

2.	The task selection process seems like a language-agnostic process. Language similarity can be easily replaced by other similarity metrics. Is there any specific reason that you apply the method to the cross-lingual scenario? Can the method be applied to other monolingual few-shot knowledge transfer tasks?


**Reasons To Accept:**

1.	The method is intuitive and effective. Meanwhile, the paper writing is clear.
2.	Experiments are comprehensive and support the claims of the paper.


**Reasons To Reject:**

1.  I didn’t see any obvious weakness in this paper.



**Reproducibility:**

4: Could mostly reproduce the results, but there may be some variation because of sample variance or minor variations in their interpretation of the protocol or method.

**Reviewer Confidence:**

3: Pretty sure, but there's a chance I missed something. Although I have a good feel for this area in general, I did not carefully check the paper's details, e.g., the math, experimental design, or novelty.

---

> ### Author Rebuttal · Authors · 2023-08-28
>
> Thank you very much for dedicating your time to reviewing our paper and providing your professional evaluation. Here are my responses to the questions you raised:
>
> - **Q1: About the reason to evaluate the optimized model on the target language test dataset directly.**
>
> You understand correctly. MAML indeed comprises two stages: the first stage involves meta-learning training, and the second stage involves adaptation learning on new meta-tasks. We did not perform the adaptation learning stage because our primary focus was on evaluating the few-shot learning capability after data selection. This capability can be assessed after training in the first stage. As seen from the results in X-METRA-ADA, the complete two-stage learning process leads to better performance. However, this was not the primary focus of our study.
>
> - **Q2: About the particularity of data selection strategy design in cross-language scenarios.**
>
> Semantic and task-based similarity data selection strategies are language-agnostic and find application in various scenarios, as mentioned in our related work. Therefore, the main focus of this paper lies in choosing more suitable data based on syntactic distance measurement. Lines 81 to 98 elaborate on why we opt for syntactic similarity sampling in cross-lingual scenarios. For short, due to the existence of syntactic differences between languages, we believe that selecting samples with more similar syntactic features will be more beneficial for cross-lingual transfer. Perhaps this idea can also be generalized to knowledge transfer tasks that share structural differences as a distinguishing feature.
>
> I hope I have effectively answered your questions. If you have any other specific suggestions or requests, we would be more than happy to consider and address them appropriately. Once again, thank you for your valuable feedback.

---

### Official Review · Reviewer_CVtE · 2023-08-05

**Soundness:** 3

**Excitement:**

3: Ambivalent: It has merits (e.g., it reports state-of-the-art results, the idea is nice), but there are key weaknesses (e.g., it describes incremental work), and it can significantly benefit from another round of revision. However, I won't object to accepting it if my co-reviewers champion it.

**Paper Topic And Main Contributions:**

The work explores model agnostic meta learning (MAML) for cross-lingual learning, focusing on data selection strategies that construct the meta tasks for MAML. Data selection strategies include semantic similarity (previous work) and a newly proposed syntactic similarity metric model. Experiments on TydiQA (QA) and WikiANN (NER) show improved results for the proposed approach.

**Reasons To Accept:**

* The work is well situated within previous work, with improvement using syntactic distance based rather than semantic one
* Experiments compare two models (mBERT & XLMR), two datasets and multiple baselines and data selection strategies

**Reasons To Reject:**

* The method is quite elaborate and there is no mention of releasing the paper code which will make replication difficult
* Authors compare two encoder models mBERT and XLMR, perhaps more interesting would have been replacing mBERT eg with a seq2seq model like mT5, [1] shows good results with mT5 (79.5 with full training and 69.4 with translate train approach, compare to best reported results by authors 65.6 which uses 10% of other language as DEV)

[1] ByT5: Towards a Token-Free Future with Pre-trained Byte-to-Byte Models. https://arxiv.org/pdf/2105.13626v3.pdf

**Reproducibility:**

5: Could easily reproduce the results.

**Reviewer Confidence:**

3: Pretty sure, but there's a chance I missed something. Although I have a good feel for this area in general, I did not carefully check the paper's details, e.g., the math, experimental design, or novelty.

**Typos Grammar Style And Presentation Improvements:**

* line 58: "models trained with randomly 056 sampled meta-tasks (X-METRAseed ) generally per- 057 form either better or worse than models trained 058 with meta-tasks sampled by semantic similarity" is quite confusing, perhaps rephrase into your main point that there is a big variation of results according to the selection and hence better selection can generate better results.

* 94: Figure(b) -> Figure 1(b)

Figure 1: X-METRA1 results are taken from Mhamdi (Table 8, X-METRA block), though results don't seem to match: bn 39 vs 38 (Mhandi vs this paper), fi 59.1 vs 56.6, id 65.2 vs 63.8 (ar matches)

* 108: "to adopt data select strategy" -> as a data selection strategy

* 138: "In recent X-METRA-ADA M’hamdi 138 et al. (2021), it follows the setting of cross-lingual" sentence needs improvement

* 178: "for the inner step as Equation 2" -> "for the inner step in Equation 2"

* 183: "The optimized-based meta-learning algorithm can perform the model fast adaptation to a new task." ?

* 261: "randomly choose q query" -> choosing

* 263: "draw the k/q" k is not defined?

Table 1: TydiQA-GP -> TydiQA-GoldP to align with previous work notation

* 440: "For WikiAnn, we added the competitive and challenging zero-shot baselines with pseudo- labeled data, including CROP(Yang et al., 2022) and SL_LEU(Xu et al., 2021)" it is not clear why these baselines are added? Perhaps also add the more competitive translate-train baseline from ByT5 (which is better than the reported results in the paper)

* Figure 3: "The effect gain", do you mean the absolute gain?

* 616: "There are more work will need to be done on the syntactic differences and syntactic distance metric." rephrase

---

> ### Author Rebuttal · Authors · 2023-08-28
>
> Thank you very much for your thorough reading of our paper and your professional assessment. We will revise our expression based on your detailed suggestions to make it more reader-friendly. Below is our response to the points you raised:
>
> - **R1: Regarding the code.**
>
> We've already provided the code as supplementary materials with the submission and will publicly release the code to facilitate reproducing our results.
>
> - **R2: About replacing PLM with a seq2seq model like mT5 and the strong translation-based baselines.**
>
> Our method is adaptable to various types of pre-trained models, including seq2seq models like mT5, with only minor code modifications required. The models mentioned in the reference you provided fine-tuned on all target languages data or translation data, which is significantly more than the DEV data (10\% of training data in one target language) we used. The result 69.4 or 79.3 is obtained by a mT5\_XXL model with 13 billion parameters, much larger than our model with 550 million parameters based on XLM-R\_large.
>
> If we compared with ByT5 based on the mT5\_base model (580M parameter) in the translate-train setting, we can have improvement by 2.7 and 3.9 on the F1 and EM metric of TydiQA-GoldP. Furthermore, the results reported in the XTREME benchmark for the Translate-train baseline are also lower than our model in the TydiQA-GoldP task. Regarding the strong translation-based baselines, intuitively, they have access to more target language training data than we do. We will put the results of this baseline in the appendix to supplementally demonstrate the advantages of our model in the few-shot scenario.
>
> Moreover, your suggestion has prompted us to consider expanding our application to include NLG tasks to further validate the effectiveness of our approach.
>
> - **About Typos Grammar Style And Presentation Improvements:**
>
> Thank you for your careful reading and patient revisions. We will also make thorough revisions to our new draft.
>
> (1) Thank you for your suggested corrections to Line 58, which have made the sentence clearer and more on-topic.
>
> (2) In Figure 1, except for the 'ar' result, which is the correct source, all others were incorrectly excerpted as results of the FT method. We have corrected this in the new version.
>
> (3) Line 183: MAML consists of two stages, meta-learning, and adaptation. So, what Line 183 intended to convey is that after the optimization by meta-learning step, the model can adapt more quickly to new meta-learning tasks.
>
> (4) **The reason for choosing the CROP and SL\_LEU models** is that they share the same pre-trained model baseline with us, and they use only monolingual data from a single target language to generate pseudo-labels. As for scenarios like ByT5 where all target language training data is used, SL\_LEU also provides comparisons, and the results surpass this setting.
> Although ensuring fairness is challenging compared to our use of a small amount of labeled data, these models do provide a strong baseline.
>
> (5) $\delta$ represents the gain of our method relative to the FT method, which falls into the category of relative gains. In the new version, we will formalize $\delta=\frac{MeTaCo-XMT\ -\ FT}{FT}$.
>
> Thank you again for your recognition of our work and for pointing out our deficiencies. Looking forward to further communication.

---

### Official Review · Reviewer_cvjj · 2023-08-14

**Soundness:** 3

**Excitement:**

3: Ambivalent: It has merits (e.g., it reports state-of-the-art results, the idea is nice), but there are key weaknesses (e.g., it describes incremental work), and it can significantly benefit from another round of revision. However, I won't object to accepting it if my co-reviewers champion it.

**Paper Topic And Main Contributions:**

This paper proposes MeTaCo-XMT, a meta-task collector-based cross-lingual meta-transfer framework for data selection in meta-transfer learning. The paper also propose a syntactic distance metric model consisting of a syntactic encoder block based on pretrained models and a distance metric block using word move’s distance. They conduct experiments with different data selection strategies to validate the framework on WikiAnn and TydiQA.

**Questions For The Authors:**

- Although the framework is proposed as a data selector, multilingual NLP has other challenges such as data availability and quality, as pointed out in Yu et al. (2022) https://arxiv.org/abs/2211.15649. Can the proposed framework tackle these challenges? Why would we use this framework rather than collected suggestions?
- What is $x$, $x^d$ and $x^c$? Is it the same as in the WMD paper? There are a lot of different notations so maybe explain before using them?

**Reasons To Accept:**

Interesting framework and metric! The metric focuses on syntactical distance rather than semantic.

**Reasons To Reject:**

- While the experiment is conducted on both NER and MRC, Section 6 almost only mentioned NER task, why is that? I would love to see more in-depth analysis regarding the framework on other tasks in Section 6.
- Presentation should be improved. The authors use many space to describe the method and implementation, yet the analysis is not enough to convince me that the proposed framework is useful.
- Many bibliography reference style errors.

**Reproducibility:**

4: Could mostly reproduce the results, but there may be some variation because of sample variance or minor variations in their interpretation of the protocol or method.

**Reviewer Confidence:**

4: Quite sure. I tried to check the important points carefully. It's unlikely, though conceivable, that I missed something that should affect my ratings.

**Typos Grammar Style And Presentation Improvements:**

- Bibliography style is incorrect and inconsistent, such as line 135, 138, 152 and 156. If it is the beginning of the sentence, you should use Kumar et al. (2022) instead of (Kumar et al. 2022).

---

> ### Author Rebuttal · Authors · 2023-08-29
>
> We extend our heartfelt thanks to you for your valuable insights and thoughtful assessment of our work. Below is my response.
>
> - **R1: About more in-depth analysis regarding the framework on other tasks.**
>
> NER and MRC represent token-level and passage-level comprehension tasks, respectively. We employ these two tasks of different levels to validate the effectiveness of our method. However, the NER task aligns more closely with our syntactic similarity selection strategy, as it is the base for syntactic analysis tasks. Therefore, due to space limitations, we have chosen to conduct a detailed analysis specifically on the NER task.
>
> We are also conducting analytical experiments on the MRC data, which will be included in the appendix of the new version. The following are some experimental results completed so far, mainly about the results of different $NUM$ settings on low- (bn) and extremely low-resource languages (sw, te) in the TydiQA-GP dataset. It can be seen from the experimental results that when the target language only has 20 annotation instances, the semantic similarity selection is not even as good as the random selection method. The selection based on task similarity still has improvement over the random scheme. Our proposed MeTaCo-XMT model based on the syntactic distance measure can always maintain the advantage in the few-shot learning setting.
>
> |     | **model**      | **bn**                  | **sw**                  | **te**                  | **avg**         |
> | --- | -------------- | ----------------------- | ----------------------- | ----------------------- | --------------- |
> | NUM | mBert baseline | 35.2 / 51.8             | 38.4 / 47.8             | 39.9 / 59.8             | 37.8 / 53.1     |
> | 30  | FT             | 36.2±3.9 / 49.8±3.0     | 41.5±1.0 / 58.4±1.4     | 47.0±1.7 / 58.2±2.2     | 41.6 / 55.5     |
> |     | random         | 39.6±1.4 / 54.0±1.3     | 38.3±1.4 / 57.0±0.6     | 50.5±1.7 / 62.8±1.5     | 42.8 / 57.9     |
> |     | sem            | 37.8±1.1 / 52.5±0.5     | 38.7±0.1 / 57.2±0.1     | 50.2±1.0 / 62.7±0.9     | 42.2 / 57.5     |
> |     | task           | 38.9±0.6 / 56.2±0.7     | 43.6±0.6 / 56.7±0.7     | 51.2±0.2 / 63.4±0.4     | 44.6 / 58.8     |
> |     | syn            | **41.2±1.1 / 56.1±0.8** | **47.6±0.5 / 60.4±0.8** | **53.2±0.4 / 65.3±0.3** | **47.6 / 60.6** |
> | 20  | FT             | 34.9±2.7 / 47.6±1.9     | 39.3±2.1 /58.6±0.6      | 42.1±3.0 / 53.9±3.8     | 38.8 / 53.4     |
> |     | random         | 33.8±2.5 / 48.1±2.4     | 39.0±1.8 /57.6±2.2      | 46.5±1.0 / 60.2±1.2     | 39.8 / 55.3     |
> |     | sem            | 34.2±2.3 / 47.8±1.5     | 38.2±1.3 / 56.2±1.0     | 45.8±1.8 / 60.3±0.5     | 39.4 / 54.8     |
> |     | task           | 36.3±1.5 / 48.8±1.2     | 40.4±0.7 / 58.8±0.6     | 46.8±0.9 / 58.7±0.4     | 41.2 / 55.4     |
> |     | syn            | **37.8±0.7 / 54.6±0.8** | **42.8±0.6 / 59.4±0.5** | **48.8±0.8 / 61.3±0.3** | **43.1 / 58.4** |
>
> - **R2R3: About presentation and more analysis.**
>
> We truly value your insights, which are vital for enhancing our paper's quality. Our comprehensive method description is intended to aid readers in understanding our motivations and the experimental setup. We have also included additional experiments and analysis in Appendix E to further showcase our method's performance.
>
> We understand your concern that an excessive amount of method and implementation details might divert readers' attention and affect the comprehensibility of the experimental analysis. In the new version, we will simplify the description of our methods and implementation, placing more space on experiments and analysis. This adjustment aims to ensure that readers can more easily comprehend our work and be persuaded by our experimental results.
>
> Additionally, we appreciate you pointing out the errors in our reference formatting, and we will address these issues one by one.
>
> - **Q1: About challenges of data that our approach can address in multilingual NLP**
>
> While our proposed framework primarily serves as a data selector, it indirectly contributes to addressing these challenges of data by:
>
> (1)Efficient Resource Utilization: Our framework can maximize the utility of available multilingual data, as shown in Table 10, we can use only 5 target labeled data to obtain great improvement in extremely-low resource languages NER task, thereby we can mitigate data scarcity issues to some extent.
>
> (2)Enhanced Data Quality: By selecting data with similar syntactic characteristics, our framework can potentially improve data quality, as it is more likely to contain relevant and high-quality examples.
>
> While our framework offers a solution, further research and improvements are ongoing to enhance its effectiveness in tackling challenges related to multilingual data availability and quality, such as using translated English Datasets. Although it is difficult to obtain high-quality translation models for low-resource languages, combining our method may also be a new way to solve this difficulty. Hope for your further discussion.
>
> - **Q2: About the explanation of notations**
>
> I apologize for any confusion caused by the organization of our text. In lines 281-282, we introduce these symbols, which represent three parallel sentences: the source language sentence, the distant language sentence, and the close language sentence. Explanations for these superscripts are also provided in Figure 2. It's possible that the explanations for identical characters were too distant in the text, and we will make appropriate adjustments in the new version. We will also check and correct the descriptions of other notations one by one.
>
> Thank you again and look forward to further discussions.

---

### Meta-Review · Area_Chair_ATNL · 2023-09-19

**Recommendation:** 3

**Metareview:**

This paper proposes MeTaCo-XMT, a framework based on model agnostic meta-learning (MAML) for cross-lingual learning, focusing on data selection strategies that construct the meta tasks for MAML. Overall, the empirical results in the paper show that MeTaCo-XMT has better results than comparable frameworks, and some interesting takeaways are that using syntactic distance is important rather than semantic one. However, some questions on the limitations of experimenting only on mBERT and XLMR, and the data availability for each language on different tasks.  Based on the reviews, the AC acknowledges that the analysis and results across multiple tasks are interesting, but recommends improvement in presenting the results, especially in resolving the extremely small fonts used in multiple results tables.

---

### Decision · Program_Chairs · 2023-10-07

**Decision:**

Accept-Findings

**Comment:**

This paper proposes MeTaCo-XMT, a framework based on model agnostic meta-learning (MAML) for cross-lingual learning, focusing on data selection strategies that construct the meta tasks for MAML. Overall, the empirical results in the paper show that MeTaCo-XMT has better results than comparable frameworks, and some interesting takeaways are that using syntactic distance is important rather than semantic one. However, some questions on the limitations of experimenting only on mBERT and XLMR, and the data availability for each language on different tasks.  Based on the reviews, the AC acknowledges that the analysis and results across multiple tasks are interesting, but recommends improvement in presenting the results, especially in resolving the extremely small fonts used in multiple results tables.